# Susceptibility of Meropenem-Resistant and/or Carbapenemase-Producing Clinical Isolates of *Enterobacterales* (*Enterobacteriaceae*) and *Pseudomonas aeruginosa* to Ceftazidime-Avibactam and Ceftolozane-Tazobactam as Assessed by In Vitro Testing Methods

**DOI:** 10.3390/antibiotics11081023

**Published:** 2022-07-29

**Authors:** Venere Cortazzo, Brunella Posteraro, Giulia Menchinelli, Flora Marzia Liotti, Tiziana D’Inzeo, Barbara Fiori, Francesco Luzzaro, Maurizio Sanguinetti, Teresa Spanu

**Affiliations:** 1Dipartimento di Scienze Biotecnologiche di Base, Cliniche Intensivologiche e Perioperatorie, Università Cattolica del Sacro Cuore, 00168 Roma, Italy; cortazzo.venere@gmail.com (V.C.); brunella.posteraro@unicatt.it (B.P.); tiziana.dinzeo@unicatt.it (T.D.); teresa.spanu@policlinicogemelli.it (T.S.); 2Dipartimento di Scienze Mediche e Chirurgiche, Fondazione Policlinico Universitario A. Gemelli IRCCS, 00168 Roma, Italy; 3Dipartimento di Scienze di Laboratorio e Infettivologiche, Fondazione Policlinico Universitario A. Gemelli IRCCS, 00168 Roma, Italy; giulia.menchinelli@policlinicogemelli.it (G.M.); floramarzia.liotti@policlinicogemelli.it (F.M.L.); barbara.fiori@policlinicogemelli.it (B.F.); 4Azienda Ospedaliera A. Manzoni, 23900 Lecco, Italy; f.luzzaro@asst-lecco.it

**Keywords:** susceptibility testing, ceftazidime-avibactam, ceftolozane-tazobactam, VITEK 2, ETEST, broth microdilution method, meropenem-resistant Gram-negative isolates

## Abstract

This study aimed to assess the comparability of in vitro susceptibility testing methods to ceftazidime-avibactam (CZA) and ceftolozane-tazobactam (C/T). Meropenem-resistant and/or carbapenemase-producing clinical isolates of *Enterobacterales* (*Enterobacteriaceae*) and *Pseudomonas aeruginosa* were tested by both bioMérieux ETEST and VITEK-2 AST-N397 card and compared with a Micronaut AST-system broth microdilution (BMD) method. CZA and C/T MICs were interpreted using EUCAST breakpoints. Of the 153 *Enterobacteriaceae* isolates, 55.6% and 0.0% (VITEK 2) and 56.9% and 0.0% (ETEST and BMD) were susceptible to CZA and C/T, respectively. Of 52 *P. aeruginosa* isolates, 50.0% and 40.4% (VITEK 2, ETEST, and BMD) were susceptible to CZA and C/T, respectively. The essential agreement (EA) was 96.1% (197/205; VITEK 2 versus BMD) and 95.6% (196/205; ETEST versus BMD) for CZA testing, whereas EA was 98.0% (201/205; VITEK 2 versus BMD) and 96.6% (198/205; ETEST versus BMD) for C/T testing. The categorical agreement (CA) was 98.0% (201/205; VITEK 2 versus BMD) and 100% (ETEST versus BMD) for CZA testing, whereas CA was 100% (VITEK 2 versus BMD) and 100% (ETEST versus BMD) for C/T testing. Categorical errors regarded four *Enterobacteriaceae* isolates. VITEK 2 and ETEST yielded equivalent CZA and C/T susceptibility testing results, compared to the BMD method, in such a clinical context.

## 1. Introduction

Specifically developed to target carbapenem-resistant and/or multidrug-resistant (MDR) Gram-negative bacteria, ceftazidime/avibactam (CZA), and ceftolozane/tazobactam (C/T) are β-lactam/β-lactamase inhibitor combination antibiotics [1], in which a well-known third-generation cephalosporin (i.e., ceftazidime) couples with a novel non-β-lactam β-lactamase inhibitor (i.e., avibactam) and vice versa [2]. Whereas ceftolozane prevents the AmpC β-lactamase-mediated β-lactam hydrolysis providing greater steric hindrance than ceftazidime, avibactam (a diazabicyclooctane non-β-lactam inhibitor) restores the activity of β-lactams against Gram-negative bacteria. These include organisms producing class A, C, and D β-lactamases (i.e., extended-spectrum β-lactamases (ESBLs), AmpC β-lactamases, or carbapenemases) that are resistant to tazobactam (a sulfone β-lactam inhibitor) [2]. Among metallo-β-lactamase (MBL)-type class B carbapenemases, IMP-1, NDM-1, and VIM-1 are not inhibited by both tazobactam and avibactam [2]. Unless these exceptions, CZA and C/T are effective not only against carbapenem-resistant *Enterobacterales* (*Enterobacteriaceae*) or *Pseudomonas aeruginosa* isolates but also against MDR *P. aeruginosa* isolates, in which porin loss (a major MDR cause together with active efflux in this species) [3] does not affect the ceftolozane effectiveness while significantly increases the in vitro MICs of ceftazidime. From a therapeutic standpoint [4], the two combination antibiotics are particularly helpful in the treatment of complicated intra-abdominal and complicated urinary tract infections as previously defined by the U.S. Food and Drug Administration (FDA).

In keeping with an ever-increasing demand for clinical microbiology laboratories to perform in vitro antimicrobial susceptibility testing (AST) for CZA and C/T in clinically relevant Gram-negative bacteria [5,6], it has become essential to assess the comparability of available AST methods for these antibiotics. To our knowledge, only three European studies (two from Germany and one from France) have recently reported on this issue [7,8,9], with one of them including the VITEK 2 AST-XN12 card (bioMérieux, Marcy l’Étoile, France; this card includes CZA and C/T in the panel of routinely tested antimicrobial agents) among assessed AST methods [8]. Consistently, studies used EUCAST breakpoints to categorize isolates from their collections of *Enterobacterales* (*Enterobacteriaceae*) and/or *P. aeruginosa* against CZA alone [8,9] or those of *P. aeruginosa* against CZA and C/T concomitantly [7]. Here, our isolates’ collection was tailored to include meropenem-resistant and/or carbapenemase-producing *Enterobacteriaceae* and *P. aeruginosa* organisms. With this collection, we assessed the abilities of VITEK 2 AST-N397 card (the version currently marketed by bioMérieux in Italy) and ETEST (bioMérieux) to yield equivalent results for CZA or C/T susceptible/resistant isolates. Using the Micronaut AST system (Merlin Diagnostika, Bornheim, Germany) based broth microdilution (BMD) method as the reference method, agreement, and error rates of VITEK and ETEST results, respectively, were calculated.

## 2. Results

We studied a previously characterized collection of *Enterobacteriaceae* (*n* = 153, mostly *Klebsiella pneumoniae* and *Escherichia coli*) or *P. aeruginosa* (*n* = 52) clinical isolates (Table 1) for the in vitro susceptibility to CZA and C/T combination antibiotics. In total, 159 (77.6%) of 205 (168 meropenem resistant and 37 meropenem susceptible) isolates harbored at least one carbapenem (meropenem) resistance-conferring gene that encodes for an IMP–, KPC–, NDM–, OXA-48-like–, or VIM–type carbapenemase [3]. As detailed in Table 1, the VIM-1 carbapenemase gene was found in 15 (41.7%) of 36 meropenem-susceptible *Enterobacteriaceae* isolates and in one (100%) of the only meropenem-susceptible *P. aeruginosa* isolate. Among those detected in meropenem-resistant isolates (122/168, 72.6%), KPC-3 (67/128, 52.3%), NDM-1 (27/128, 21.1%), and VIM-1 (18/128, 14.1%) were the most frequent carbapenemase genes. Non-carbapenemase associated meropenem-resistance mechanisms remained unexplored in remaining isolates (46/168, 27.4%). However, these isolates were all *P. aeruginosa* organisms with (not shown) MDR or extensively drug-resistant (XDR) phenotypes [10], which could be attributed to multiple antimicrobial resistance mechanisms (active efflux, porin alteration, or deficiencies) [3].

Applying EUCAST breakpoints (Appendix A), the proportion of *Enterobacteriaceae* or *P. aeruginosa* isolates susceptible to CZA was similar by Micronaut AST system (55.1%, 113/205 isolates), VITEK 2 (54.1%, 111/205 isolates), and ETEST (55.1%, 113/205 isolates). Likewise, the proportion of *Enterobacteriaceae* or *P. aeruginosa* isolates susceptible to C/T was 10.2% (21/205 isolates [only *P. aeruginosa* organisms]) by all three methods. According to the presence of carbapenemases with known resistance to the tazobactam-mediated inhibition [2], none of the 153 *Enterobacteriaceae* isolates was susceptible to C/T. As shown in Appendix A, MIC_90_ values for CZA or C/T combination antibiotic were similar when tested by the three methods. Otherwise, the MIC_50_ values for CZA were lower than for C/T when tested by Micronaut AST system (8/4 µg/mL versus >64/4 µg/mL), VITEK 2 (8/4 µg/mL versus ≥32/4 µg/mL), or ETEST (4/4 µg/mL versus >256/4 µg/mL) methods. According to the presence/expression of carbapenemase genes (Table 1), 67 (*Enterobacteriaceae*, *n =* 61; *P. aeruginosa*, *n =* 6), isolates resistant to both CZA and C/T (Appendix A) harbored a class B (MBL) carbapenemase gene (*bla*_IMP-1_, *bla*_NDM-1_, *bla*_NDM-5_, *bla*_VIM-1,_ or *bla*_VIM-4_) alone (62 isolates) or in combination with a non-MBL carbapenemase gene (5 isolates). Likewise, 87 of 92 *Enterobacteriaceae* isolates harboring a class A (*bla*_KPC-2_ or *bla*_KPC-3_) and/or class D (*bla*_OXA-48_) carbapenemase gene were found to be susceptible to CZA (Appendix A). This raised the (unexplored) possibility that a CZA resistance-conferring mutation or unspecified mechanism affecting the carbapenemase (*bla*_KPC-3_) gene [11,12,13] has occurred in 5 (7.5%) of 67 *bla*_KPC-3_ harboring *Enterobacteriaceae* isolates studied by us.

We purposely tested *Enterobacteriaceae* or *P. aeruginosa* isolates for which the probability of being resistant to CZA and/or C/T was relatively high [14,15,16]. Indeed, our reliability assessment of (VITEK 2 AST-N397 card or ETEST) AST methods used *Enterobacteriaceae* (66/153 and 153/153) or *P. aeruginosa* (26/52 and 31/52) isolates whose Micronaut AST MICs to CZA and C/T, respectively, fall within the EUCAST resistant category (CZA MICs, >8/4 µg/mL; C/T MICs, >2/4 µg/mL for *Enterobacteriaceae* and >4/4 µg/mL for *P. aeruginosa*). At initial testing (Appendix A), 19 (10 *K. pneumoniae* and 9 *P. aeruginosa*) isolates had VITEK 2/ETEST category (susceptible/resistant) results that did not agree with those of the Micronaut AST method. Repeat testing of these isolates allowed us to resolve 15 of 19 discrepancies, resulting into only four categorical errors that regarded *K. pneumoniae* isolates when tested with the VITEK 2 AST-N397 card. Three (two *bla*_KPC-2_ and one *bla*_KPC-3_ harboring) isolates were categorized as (false) resistant (MICs, ≥16 µg/mL) and one (*bla*_NDM-1_ harboring) isolate was categorized as (false) susceptible (MIC, 8 µg/mL) to CZA (Table 2).

Overall, there were 98.0% (201/205) of categorical agreement (CA) results for CZA by the VITEK 2 method as well as 100% (205/205) of CA results for CZA by the ETEST method or for C/T by both the methods (Table 2). As detailed in Figure 1, VITEK 2 or ETEST methods generated MICs that differed from Micronaut AST MICs, respectively, up to 3- or 5-fold for CZA and up to 2- or 6-fold for C/T. However, differences went unnoticed by CA in 24 of 28 instances because the isolates’ MICs to CZA or C/T were far below the EUCAST susceptible breakpoint (6 isolates) or far above the EUCAST resistant breakpoint (18 isolates). Only involving *K. pneumoniae* isolates when tested against CZA with the VITEK 2 (AST-N397 card) method, very major error (VME) and major error (ME) rates in our study were 1.1% (1/92) and 2.7% (3/113), respectively.

## 3. Discussion

Table 3 summarizes the results from similar studies (i.e., European studies that used interpretive EUCAST breakpoints) conducted so far [7,8,9]. Like us, one study tested both CZA and C/T combination antibiotics using VITEK 2 and ETEST methods [7], whereas two other studies tested only CZA using the ETEST method [8,9].

Comparing our study with that by Daragon et al. [7] revealed an almost identical number of Gram-negative isolates (200 (albeit restricted to *P. aeruginosa*) and 205, respectively) studied in total. However, numbers/proportions of *P. aeruginosa* isolates susceptible to CZA (34 isolates, 17.0%) or C/T (40 isolates, 20.0%) based on BMD MICs were relatively lower in that study [7]. Furthermore, a consistent number of isolates had CZA (and C/T) MICs near to the EUCAST breakpoints (±1-log_2_); thus, in part, accounting for 18.1% (30/166) of false-resistant isolates (MEs) or 8.8% (3/34) of false-susceptible isolates (VMEs) when tested for CZA with the VITEK 2 (AST-XN12 card) method [7]. Otherwise, CZA testing with the ETEST method on the same isolates revealed an inverse situation, being the ME rate as low as 3.0% (5/166 isolates) and the VME rate as high as 14.7% (5/34 isolates). These error rates were reminiscent of those reported by Schaumburg et al. [8] for a clinical collection of MDR/XDR *P. aeruginosa* isolates tested against CZA (Table 3) but were dissimilar from the error rates found by Kresken et al. [9], which were equal to (ME) or a very little greater than (VME) zero. If looking at the Kresken et al. [9] study’s findings for *Enterobacterales* (*Enterobacteriaceae*) and *P. aeruginosa* isolates at a whole (200 isolates, of which 34 were CZA-resistant), we noticed them to be consistent with our study’s findings (205 isolates, of which 92 were CZA-resistant). Since 2018 (the year the Kresken et al.’s study was published), until the present study (Table 3), the number of Gram-negative bacterial isolates with CZA or C/T MICs higher than the EUCAST susceptible breakpoint (if those studied by the authors [9] were from a clinical collection) appears to be substantially increased. However, the possibility raised by Daragon et al. [7] to overestimate the CZA resistance when testing clinical *P. aeruginosa* isolates with the VITEK 2 AST-XN12 card cannot be excluded based on the findings from our study.

Confirming independent U.S. multicenter evaluation findings [17,18], in a much more challenging set of isolates, we have shown that the performance of the VITEK 2 AST-N397 card to determine the susceptibility of clinically relevant Gram-negative bacterial species to CZA and C/T combination antibiotics was comparable to that of the ETEST method. However, performance assessment of both VITEK 2 and ETEST methods in our study might be biased because we used a comparison method (i.e., the Micronaut AST system) that is really a surrogate of the widely accepted, albeit less practicable, BMD reference method [19]. Otherwise, final assessment of VITEK 2/ETEST MIC results implied repeat testing of isolates with initial CA discrepancies. Knowing the reasons behind these discrepancies could make the occurrence of categorical errors with both VITEK 2 and ETEST methods very negligible in the future. Finally, our study was intrinsically limited by the relatively small number of Gram-negative bacterial isolates tested.

## 4. Materials and Methods

### 4.1. Bacterial Isolates

Two hundred and five non-duplicate Gram-negative (153 *Enterobacteriaceae* and 52 *P. aeruginosa*) isolates from two Italian hospital clinical microbiology laboratories’ collections (January 2015 through December 2020) were available for inclusion in this study. Of these isolates, 110 (53.7%), 64 (31.2%), and 31 (15.1%) were recovered from bloodstream, respiratory tract, or urinary tract infections, respectively. As detailed in Table 1, isolates had been characterized as having meropenem-resistant phenotypes (168 isolates) and/or carbapenemase genes (165 isolates, of which 37 were meropenem susceptible). In these isolates, detection of carbapenem-resistance associated genes (*bla*_IMP-1_, *bla*_KPC-2_, *bla*_KPC-3_, *bla*_NDM-1_, *bla*_NDM-5_, *bla*_OXA-48_, *bla*_VIM-1_, and *bla*_VIM-4_) had been performed by PCR (using primers and thermal conditions already described [20,21]) followed by sequence analysis. Isolates were kept at −80 °C in 20% glycerol and sub-cultured prior to use on tryptic soy agar with 5% sheep’s blood. Quality control (QC) testing with *K. pneumoniae* ATCC 700603, *E. coli* ATCC 35218, and *P. aeruginosa* ATCC 27853 was performed each day clinical isolates were tested, and values were always within the acceptable CLSI range for QC isolates [19].

### 4.2. Antimicrobial Susceptibility Testing

According to the respective manufacturer’s instructions, Micronaut AST system-based BMD, VITEK 2 AST-N397 card, and ETEST were used to determine MICs of ceftazidime (from ≤1 to >64 µg/mL, ≤0.12 to ≥16 µg/mL, and ≤0.016 to >256 µg/mL, respectively) with 4 µg/mL avibactam. Similarly, MICs of ceftolozane (from ≤0.5 to >64 µg/mL, ≤0.25 to ≥32 µg/mL, and ≤0.016 to >256 µg/mL, respectively) with 4 µg/mL tazobactam were determined. Each isolate was tested in parallel by the BMD, ETEST, and VITEK 2 methods, the same day, using a bacterial suspension that was adjusted at 0.5 McFarland in 0.9% NaCl solution for both BMD and ETEST methods or in a specific 0.45% saline solution provided by bioMérieux for the VITEK 2 method. EUCAST (version 11.0, 2021) clinical breakpoints [22] for *Enterobacterales* (*Enterobacteriaceae*) or *P. aeruginosa* were used to interpret MICs to CZA and C/T, respectively.

### 4.3. Agreement Analysis

Using the Micronaut AST system-based BMD as the reference method, essential agreement (EA; defined as MIC results differing by a maximum of 1-log_2_ concentration), CA (defined as MIC results within the same susceptibility category), ME (defined as false resistance), and VME (defined as false susceptibility) assessments were performed according to standard criteria. Because the (BMD) reference method used in this study does not contain ceftazidime and ceftolozane concentrations lower/higher than 1/64 and 0.5/64 µg/mL, respectively, EA calculation for the VITEK 2 or the ETEST method included both isolates with and without on-scale results (i.e., those that fall within the test range of the reference method). This was in accordance with the AST systems guidance document published by the U.S. FDA in 2009 (http://www.fda.gov/MedicalDevices/DeviceRegulationandGuidance/GuidanceDocuments/ucm080564.htm; accessed on 1 July 2022). When MIC results fell between conventional two-fold dilutions (half dilutions), ETEST results were rounded up to the next upper doubling dilution that was used with the BMD method. The above-mentioned EUCAST breakpoints allowed assessing the CA of the VITEK 2 or the ETEST method with BMD as well as the error rates for both VITEK 2 and ETEST methods. Isolates with VME or ME were retested with the VITEK 2 or the ETEST method, as appropriate, and repeat testing results were used for analysis (Appendix A).

## 5. Conclusions

Our analysis of CZA and C/T MIC results of clinically relevant *Enterobacteriaceae* or *P. aeruginosa* organisms allowed us to show that CA was excellent (98%) for the VITEK 2 AST-N397 card and complete (100%) for the ETEST method. While both ETEST and VITEK 2 allow testing the susceptibility to CZA and C/T with less hands-on time compared to reference method, the automation brought by VITEK 2 also allows its use in a routine workflow. ETEST remains an accurate alternative when these drugs are not yet available in VITEK 2 AST cards.

## Figures and Tables

**Figure 1 antibiotics-11-01023-f001:**
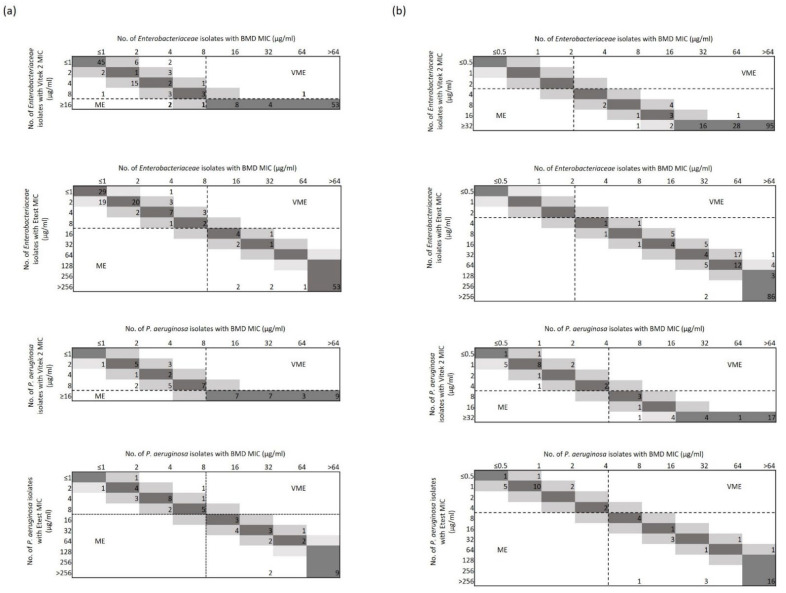
The correlation between ceftazidime/avibactam (**a**) and ceftolozane/tazobactam (**b**) MICs (expressed as µg/mL) obtained by testing *Enterobacteriaceae* (*n* = 153) and *P. aeruginosa* (*n* = 52) isolates with the BMD (reference), VITEK 2, or ETEST methods, respectively. Darker and lighter grey squares indicate, respectively, the number of isolates with a VITEK 2 or ETEST MIC corresponding to the BMD MIC and 1-log_2_ concentration for each of two antimicrobial agents. Numbers outside the squares indicate the isolates with VITEK 2 or ETEST MICs that differed by more than 1-log_2_ concentration from the BMD MIC, which determined essential agreement rates below 100% for ceftazidime/avibactam or ceftolozane/tazobactam, respectively. Dashed lines delimitate VME or ME areas to represent, respectively, the isolates falsely categorized as susceptible or resistant according to EUCAST clinical breakpoints.

**Table 1 antibiotics-11-01023-t001:** Phenotypic and genotypic characteristics of 205 Gram-negative bacterial isolates included in the study ^a^.

Isolates According to the Species Identified (*n*)	Meropenem-Susceptible/Resistant Isolates (*n*)	Carbapenemase-Producing Isolates According to the Carbapenem-Resistance Gene(s) Detected (*n*)								
		*bla* _IMP-1_	*bla* _KPC-2_	*bla* _KPC-3_	*bla* _NDM-1_	*bla* _NDM-5_	*bla* _OXA-48_	*bla* _VIM-1_	*bla* _VIM-4_	Total Genes ^b^
*Enterobacteriaceae* (153)	Susceptible (36)	–	6	10	1	–	2	15	2	36
	Resistant (117)	–	9	67	27	1	4	14	1	123
*Escherichia coli* (34)	Susceptible (18)	–	5	4	1	–	1	6	1	18
	Resistant (16)	–	1	–	14	1	1	–	–	17
*Klebsiella pneumoniae* (100)	Susceptible (8)	–	1	5	–	–	1	1	–	8
	Resistant (92)	–	8	67	12	–	3	7	–	97
Other species (19) ^c^	Susceptible (10)	–	–	1	–	–	–	8	1	10
	Resistant (10)	–	–	–	1	–	–	7	1	9
*Pseudomonas aeruginosa* (52)	Susceptible (1)	–	–	–	–	–	–	1	–	1
	Resistant (51)	1	–	–	–	–	–	4	–	5
Total species (205)	Susceptible (37)	–	6	10	1	–	2	16	2	37
	Resistant (168)	1	9	67	27	1	4	18	1	128

^a^ Before being included, isolates had been characterized for carbapenem (meropenem) resistance/carbapenemase production using phenotypic (broth microdilution)/immunochromatographic methods followed by PCR-sequencing analysis, which confirmed the presence of carbapenem-resistance (i.e., carbapenemase) gene(s) in all (100%) of the 117 meropenem-resistant *Enterobacteriaceae* isolates and in 5 (9.8%) of 51 meropenem-resistant *P. aeruginosa* isolates. The same analysis allowed to detect a carbapenem-resistance (i.e., carbapenemase) gene in 36 (100%) of 36 meropenem-susceptible but carbapenemase-producing *Enterobacteriaceae* isolates or in one (100%) of one meropenem-susceptible but carbapenemase-producing *P. aeruginosa* isolate. Based on previous AST results (not shown), meropenem MICs (range, 0.25 to 2 µg/mL) of the so-called meropenem susceptible isolates were below the S (susceptible)/I (susceptible, increased exposure) clinical breakpoint (MIC, ≤2 µg/mL) but above the epidemiological cut-off value (MIC, >0.125 µg/mL) EUCAST-defined for meropenem, thus qualifying the isolates for carbapenemase production investigation [3]. ^b^ In 1 of 16 meropenem-resistant *E. coli* isolates, 2 carbapenem-resistance genes (i.e., *bla*_NDM-1_ and *bla*_OXA-48_) had concomitantly been detected. In 5 of 91 meropenem-resistant *K. pneumoniae* isolates, two carbapenem-resistance genes each (i.e., *bla*_KPC-3_ and *bla*_VIM-1_ [3 isolates], *bla*_KPC-3_ and *bla*_OXA-48_ [1 isolate], or *bla*_NDM-1_ and *bla*_OXA-48_ [1 isolate]) had concomitantly been detected. ^c^ Other species include *Citrobacter freundii* (1 isolate), *Enterobacter cloacae* (6 isolates), *Klebsiella aerogenes* (1 isolate), *Klebsiella oxytoca* (9 isolates), *Klebsiella variicola* (1 isolate), and *Raoultella ornithinolytica* (1 isolate).

**Table 2 antibiotics-11-01023-t002:** Agreement and error rates in testing 205 Gram-negative bacterial isolates against ceftazidime/avibactam and ceftolozane/tazobactam.

Species of Isolates Tested (*n*)	Method Evaluated in Comparison with BMD	Ceftazidime/Avibactam (CZA) ^a^				Ceftolozane/Tazobactam (C/T) ^a^			
		% EA (*n*)	% CA (*n*)	% ME (*n*) ^b^	% VME (*n*) ^c^	% EA (*n*)	% CA (*n*)	% ME (*n*) ^b^	% VME (*n*) ^c^
*Enterobacteriaceae* (153)	VITEK 2	96.1 (147/153)	97.4 (149/153)	3.4 (3/87)	1.5 (1/66)	98.7 (151/153)	100 (153/153)	0 (0/0)	0 (0/153)
	ETEST	96.1 (147/153)	100 (153/153)	0 (0/87)	0 (0/66)	98.0 (150/153)	100 (153/153)	0 (0/0)	0 (0/153)
*Escherichia coli* (34)	VITEK 2	100 (34/34)	100 (34/34)	0 (0/11)	0 (0/23)	100 (34/34)	100 (34/34)	0 (0/0)	0 (0/34)
	ETEST	94.1 (32/34)	100 (34/34)	0 (0/11)	0 (0/23)	100 (34/34)	100 (34/34)	0 (0/0)	0 (0/34)
*Klebsiella pneumoniae* (100)	VITEK 2	94.0 (94/100)	96.0 (96/100)	4.0 (3/75)	4.0 (1/25)	98.0 (98/100)	100 (100/100)	0 (0/0)	0 (0/100)
	ETEST	97.0 (97/100)	100 (100/100)	0 (0/75)	0 (0/25)	97.0 (97/100)	100 (100/100)	0 (0/0)	0 (0/100)
Other species (19) ^d^	VITEK 2	100 (19/19)	100 (19/19)	0 (0/1)	0 (0/18)	100 (19/19)	100 (19/19)	0 (0/0)	0 (0/19)
	ETEST	94.7 (18/19)	100 (19/19)	0 (0/1)	0 (0/18)	100 (19/19)	100 (19/19)	0 (0/0)	0 (0/19)
*Pseudomonas aeruginosa* (52)	VITEK 2	96.2 (50/52)	100 (52/52)	0 (0/26)	0 (0/26)	96.2 (50/52)	100 (52/52)	0 (0/21)	0 (0/31)
	ETEST	94.2 (49/52)	100 (52/52)	0 (0/26)	0 (0/26)	92.3 (48/52)	100 (52/52)	0 (0/21)	0 (0/31)
Total species (205)	VITEK 2	96.1 (197/205)	98.0 (201/205)	2.7 (3/113)	1.1 (1/92)	98.0 (201/205)	100 (205/205)	0 (0/21)	0 (0/184)
	ETEST	95.6 (196/205)	100 (205/205)	0 (0/113)	0 (0/92)	96.6 (198/205)	100 (205/205)	0 (0/21)	0 (0/184)

BMD, broth microdilution; EA, essential agreement; CA, categorical agreement; ME, major error; VME, very major error. ^a^ EUCAST 2021 clinical breakpoints for CZA-susceptible (MIC, ≤8/4 µg/mL) or -resistant (MIC, >8/4 µg/mL) isolates and for CT-susceptible (MIC, ≤2/4 µg/mL for *Enterobacteriaceae*; MIC, ≤4/4 µg/mL for *P. aeruginosa*) or CT-resistant (MIC, >2/4 µg/mL for *Enterobacteriaceae*; MIC, >4/4 µg/mL for *P. aeruginosa*) isolates were used to interpret isolates’ testing results. Based on BMD testing results, all (100%) of the 153 *Enterobacteriaceae* isolates (87 CZA susceptible and 66 CZA resistant) were resistant to C/T. A total of 31 (59.6%) of 52 *P. aeruginosa* isolates (25 CZA resistant and 6 CZA susceptible) were resistant to C/T, with the remaining 1 CZA-resistant isolate being susceptible to C/T. ^b^ ME rates were calculated using the number of susceptible isolates as denominator. ^c^ VME rates were calculated using the number of resistant isolates as denominator. ^d^ Other species include *Citrobacter freundii* (1 isolate), *Enterobacter cloacae* (6 isolates), *Klebsiella aerogenes* (1 isolate), *Klebsiella oxytoca* (9 isolates), *Klebsiella variicola* (1 isolate), and *Raoultella ornithinolytica* (1 isolate).

**Table 3 antibiotics-11-01023-t003:** Studies conducted in Europe to evaluate the VITEK 2 and/or ETEST methods for determining the susceptibility of Gram-negative bacterial species to the CZA and C/T combination antibiotics.

Study ^a^	Country	Antibiotic Tested	BMD Panels (Concentration Range Tested) ^b^	Method Evaluated ^c^	Bacterial Species Tested (Source)	No. Total/no. Resistant Isolates ^d^	Method’s Performance Assessed as no. (%) of Isolates with the Indicated Result ^e^			
							EA	CA	ME	VME
This study	Italy	CZA	Micronaut AST system panels (1–64 µg/mL)	VITEK AST-N397 card	*Enterobacteriaceae*/*P. aeruginosa* (clinical isolate collection)	205/92	197 (96.1)	201 (98.0)	3 (2.7)	1 (1.1)
				ETEST			196 (95.6)	205 (100)	0 (0.0)	0 (0.0)
		C/T	Micronaut AST system panels (0.5–64 µg/mL)	VITEK AST-N397 card	*Enterobacteriaceae*/*P. aeruginosa* (clinical isolate collection)	205/184	201 (98.0)	205 (100)	0 (0.0)	0 (0.0)
				ETEST			198 (96.6)	205 (100)	0 (0.0)	0 (0.0)
Daragon, 2021 [7]	France	CZA	Thermo Fisher panels (0.5–256 µg/mL)	VITEK 2 XN12 card	*P. aeruginosa* (clinical isolate collection)	200/34	178 (89.0)	167 (83.5)	30 (18.1)	3 (8.8)
				Etest			188 (94.0)	190 (95.0)	5 (3.0)	5 (14.7)
		C/T	Thermo Fisher panels (0.5–256 µg/mL)	VITEK 2 XN12 card	*P. aeruginosa* (clinical isolate collection)	200/40	193 (96.5)	191 (95.5)	2 (1.2)	5 (12.5)
				ETEST			185 (92.5)	190 (95.0)	0 (0.0)	10 (25.0)
Schaumburg, 2019 [8]	Germany	CZA	Merlin panels (0.5–256 µg/mL)	ETEST	*P. aeruginosa* (clinical isolate collection)	192/69	182 (94.8)	181 (94.3)	6 (4.9)	5 (7.2)
Kresken, 2018 [9]	Germany	CZA	CLSI M100-S27 based panels (0.016–256 µg/mL)	ETEST	*Enterobacterales* (unspecified)	140/12	139 (99.3)	140 (100.0)	0 (0.0)	0 (0.0)
					*P. aeruginosa* (unspecified)	60/22	59 (98.3)	59 (98.3)	0 (0.0)	1 (4.5)

BMD, broth microdilution; EA, essential agreement; CA, categorical agreement; ME, major error; VME, very major error; CZA, ceftazidime/avibactam; C/T, ceftolozane/tazobactam. ^a^ Except for the present study, three other studies here listed are referred to as first author, year, and reference number. ^b^ Except for one study that used self-prepared BMD panels [9], three other studies used commercial BMD panels and, consequently, concentration ranges in these panels were less broaden. ^c^ The XN12 or AST-N397 cards used with the VITEK 2 method in two studies [7 and this study] are equivalent. ^d^ Currently available EUCAST breakpoints were used to interpret isolates’ testing results in each study. ^e^ For VITEK 2 or ETEST methods, performance was assessed in comparison with the indicated BMD method, which was used as the reference method in each study.

## Data Availability

Data may be available upon reasonable request.

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
