# Peer review of "Susceptibility of Meropenem-Resistant and/or Carbapenemase-Producing Clinical Isolates of Enterobacterales (Enterobacteriaceae) and Pseudomonas aeruginosa to Ceftazidime-Avibactam and Ceftolozane-Tazobactam as Assessed by In Vitro Testing Methods"

_antibiotics, 2022, doi:10.3390/antibiotics11081023_

Round 1

Reviewer 1 Report

This paper presents a comparison of three different in vitro testing methods (ETEST, VITEK-2 and BMD) to assess the susceptibility of clinical isolates of Enterobacteriaceae and Pseudomonas aeruginosa to the combination antibiotics CZA and C/T. Overall, the study contains interesting data. However, Introduction must be improved to include all the concepts discussed in the manuscript. The discussion of the results and conclusions must be reformulated to include only hypothesis that can be proved with the results presented in this study.  Below, I present a detailed list of my major comments/concerns.

Major points:

1-       It is not clear why the BMD method was used as the reference in this study. Instead, it is stated in the last sentence of the introduction that the three methods were compared between each other. The authors should clarify, in the introduction section, that the BMD method was used as a standard and why this is the case.

2-       In the results section, page 3, the authors affirm: “Expectedly, none of the 153 Enterobacteriaceae isolates was susceptible to C/T.” This hypothesis is not explained at all. The mechanism of action of ceftazidime, ceftolozane, avibactam and tazobactam is briefly touched in the introduction part, but the authors do not explain why it is expected that the ceftolozane/tazobactam combination should not be active against Enterobacteriaceae. This point should be clarified in the introduction or results section.

3-       Most of the bacterial isolates used in this study (153/205) are Enterobacteriaceae isolates, which are all resistant to C/T. The efficiency of using such a sample to assess the performance of the susceptibility tests should be discussed.

4-       The authors state that initial discrepancies with respect to the BMD method led to an incorrect categorization of 4 Enterobacteriaceae isolates. The authors then state that this was solved, but the reasons behind this erroneous characterization is not discussed at this point, being only briefly reported in the methods section. This should be added to the manuscript. Otherwise reporting these false categorizations is confusing.

5-       The authors claim that the results from this study vanish the hypothesis, raised in a previous study from Daragon et al., that VITEK method lead to overestimation of CZA resistance. This is, however, difficult to conclude only based on this study. The study from Daragon et al. comprises 200 P. aeruginosa isolates, in which 34 were resistant to CZA. In the present study, however, only 52 isolates were used. The conclusions are definitely different, but the authors need to explain better why they claim that they have enough evidence to say “the possibility raised by Daragon et al. [6] to overestimate the CZA resistance when testing clinical P. aeruginosa isolates with the VITEK 2 AST-XN12 card vanished based on the findings from our study”.

6-       The authors conclude that the results from this study indirectly underscore the importance of routinely testing the susceptibility for CZA and C/T antibiotics for “infections are caused by carbapenem-resistant or MDR organisms belonging to commonly isolated Gram-negative bacterial species”. However, this conclusion is not supported by the results in this study.

Minor points:

Abstract, line 13 - “Errors regarded four Enterobacteriaceae (K. pneumoniae) isolates.” This sentence is difficult to understand if only the abstract is read.

Abstract, line 14 - VITEK and ETEST yelded equivalent CZA and C/T susceptibility testing results, compared to the BMD method, in such a clinical context.

The second sentence of the Introduction section is too long and the ideas are not clear. It should be divided in, at least, two sentences in order to organize the ideas. The use of remarkably is useless in the context of this sentence.

Page 3, line 95 – “Expectedly, none of the 153 Enterobacteriaceae isolates was susceptible to C/T. Therefore, as shown (Table S1), it was not surprising that the MIC90 values for CZA or C/T combination antibiotic were similar when tested by the three methods.” What is the cause/effect relationship between these sentences? Why is therefore used here? There is no relationship between the fact that all the Enterobacteriaceae isolates are susceptible to C/T and the MIC results are similar for CZA and C/T.

Page 3, line 98 – Review the use of similarly. Again, there is no similar results between this sentence and the previous one. The use of such adverbs along the manuscript should be revised.

Categorical agreement was defined as MIC results within the same category. This definition is dubious and should be clarified.

The authors only list benefits of using ETEST or VITEK-2 tests to assess susceptibility in the conclusions section. This is important for the motivation of the study and should be, at least, briefly 

Author Response

This paper presents a comparison of three different in vitro testing methods (ETEST, VITEK-2 and BMD) to assess the susceptibility of clinical isolates of Enterobacteriaceae and Pseudomonas aeruginosa to the combination antibiotics CZA and C/T. Overall, the study contains interesting data. However, Introduction must be improved to include all the concepts discussed in the manuscript. The discussion of the results and conclusions must be reformulated to include only hypothesis that can be proved with the results presented in this study. Below, I present a detailed list of my major comments/concerns.

Answer: I am very grateful to the reviewer for appreciating my manuscript, which deals with the VITEK 2 and ETEST methods’ evaluation with respect to the relatively new β-lactam/β-lactamase inhibitor combination antibiotics. According to the comments raised by the reviewer, I modified the manuscript in the Introduction, Discussion, and Conclusions section. Below, my reported responses to the specific comments.

Major points:

  1. It is not clear why the BMD method was used as the reference in this study. Instead, it is stated in the last sentence of the introduction that the three methods were compared between each other. The authors should clarify, in the introduction section, that the BMD method was used as a standard and why this is the case.

Answer: As required, I modified the last sentence in the Introduction to clarify the issue about the reference BMD method.

  1. In the results section, page 3, the authors affirm: “Expectedly, none of the 153 Enterobacteriaceae isolates was susceptible to C/T.” This hypothesis is not explained at all. The mechanism of action of ceftazidime, ceftolozane, avibactam and tazobactam is briefly touched in the introduction part, but the authors do not explain why it is expected that the ceftolozane/tazobactam combination should not be active against Enterobacteriaceae. This point should be clarified in the introduction or results section.

Answer: I modified the Introduction and the Results sections to add some explanation about the indicated issues.

  1. Most of the bacterial isolates used in this study (153/205) are Enterobacteriaceae isolates, which are all resistant to C/T. The efficiency of using such a sample to assess the performance of the susceptibility tests should be discussed.

Answer: I added a paragraph at the end of the Discussion section to acknowledge all the limitations of the study, including the issues about the VITEK 2 and ETEST methods’ performance assessment.

  1. The authors state that initial discrepancies with respect to the BMD method led to an incorrect categorization of four Enterobacteriaceae isolates. The authors then state that this was solved, but the reason behind this erroneous characterization is not discussed at this point, being only briefly reported in the methods section. This should be added to the manuscript. Otherwise reporting these false categorizations is confusing.

Answer: I added some sentences in the Results section to comment on the incorrect categorization of some isolates tested as well as in the Discussion section to comment on the reasons underlying this incorrect categorization.

  1. The authors claim that the results from this study vanish the hypothesis, raised in a previous study from Daragon et al., that VITEK method led to overestimation of CZA resistance. This is, however, difficult to conclude only based on this study. The study from Daragon et al. comprises 200 P. aeruginosa isolates, in which 34 were resistant to CZA. In the present study, however, only 52 isolates were used. The conclusions are definitely different, but the authors need to explain better why they claim that they have enough evidence to say, “the possibility raised by Daragon et al. [6] to overestimate the CZA resistance when testing clinical P. aeruginosa isolates with the VITEK 2 AST-XN12 card vanished based on the findings from our study”.

Answer: As suggested, I modified the statement to clarify the issue and to mitigate the meaning of what were are claiming.

  1. The authors conclude that the results from this study indirectly underscore the importance of routinely testing the susceptibility for CZA and C/T antibiotics for “infections are caused by carbapenem-resistant or MDR organisms belonging to commonly isolated Gram-negative bacterial species”. However, this conclusion is not supported by the results in this study.

Answer: As suggested, I modified the Conclusions section by deleting the questionable sentence.

Minor points:

Abstract, line 13 - “Errors regarded four Enterobacteriaceae (K. pneumoniae) isolates.” This sentence is difficult to understand if only the abstract is read.

Answer: I modified the sentence to make it clearer.

Abstract, line 14 - VITEK and ETEST yielded equivalent CZA and C/T susceptibility testing results, compared to the BMD method, in such a clinical context.

Answer: I modified the sentence as suggested.

The second sentence of the Introduction section is too long, and the ideas are not clear. It should be divided in, at least, two sentences in order to organize the ideas. The use of remarkably is useless in the context of this sentence.

Answer: I modified the sentence by splitting it in two parts, as well “remarkably” was omitted.

Page 3, line 95 – “Expectedly, none of the 153 Enterobacteriaceae isolates was susceptible to C/T. Therefore, as shown (Table S1), it was not surprising that the MIC90 values for CZA or C/T combination antibiotic were similar when tested by the three methods.” What is the cause/effect relationship between these sentences? Why is therefore used here? There is no relationship between the fact that all the Enterobacteriaceae isolates are susceptible to C/T and the MIC results are similar for CZA and C/T.

Answer: I modified the three sentences to improve clarity and meaning.

Page 3, line 98 – Review the use of similarly. Again, there is no similar results between this sentence and the previous one. The use of such adverbs along the manuscript should be revised.

Answer: I modified the sentences to improve clarity and meaning, as well as I revised the use of adverbs in this and other parts of the manuscript.

Categorical agreement was defined as MIC results within the same category. This definition is dubious and should be clarified.

Answer: As suggested, I clarified the categorical agreement definition.

The authors only list benefits of using ETEST or VITEK-2 tests to assess susceptibility in the conclusions section. This is important for the motivation of the study and should be, at least, briefly.

Answer: As mentioned above, I shortened the Conclusions section to mitigate the emphasis about the VITEK 2 and ETEST methods.

Reviewer 2 Report

Essential agreement and categorical agreement should be explained in the introduction section for better understanding. 

Authors should include the primers used to amplify carbapenem-resistance gene(s) in the methods section. 

Table 1. Other species should be specified below the table. 

Revise Table 3. Some of the results are not visible. 

Figure 1 is not clear. Axis title and other information should be made visible. 

Change MIC values- mg/L to µg/ml. 

Author Response

Essential agreement and categorical agreement should be explained in the introduction section for better understanding.

Answer: I am very grateful to the reviewer for appreciating my manuscript, which deals with the VITEK 2 and ETEST methods’ evaluation with respect to the relatively new β-lactam/β-lactamase inhibitor combination antibiotics. I modified the manuscript according to the issues raised by the reviewer, particularly regarding the study limitations.

Authors should include the primers used to amplify carbapenem-resistance gene(s) in the methods section.

Answer: As required, I added details about the PCR primers and thermal cycle conditions used for the carbapenemase genes’ detection.

Table 1. Other species should be specified below the table.

Answer: A specification footnote was added as required.

Revise Table 3. Some of the results are not visible.

Answer: Table 3 was revised as required.

Figure 1 is not clear. Axis title and other information should be made visible.

Answer: I modified Figure 1 to make all information visible.

Change MIC values- mg/L to μg/ml

Answer: As suggested, I changed the MIC value expression throughout the manuscript.

Reviewer 3 Report

The issue of MDR Gram-negative bacilli has become a concern in many states in central, southern and southeastern Europe, which is why the administration of new combinations of beta-lactams / inhibitors has become necessary. The manuscript describes the results of 3 in vitro methods for testing  GNB (205 isolates) sensitivity to these new combinations.  Although it is well written, clearly presented and documented, with consistent conclusions and discussions, it would be appropriate to specify the inclusion and exclusion criteria, as well as the study limitation.

Author Response

The issue of MDR Gram-negative bacilli has become a concern in many states in central, southern and southeastern Europe, which is why the administration of new combinations of beta-lactams / inhibitors has become necessary. The manuscript describes the results of 3 in vitro methods for testing  GNB (205 isolates) sensitivity to these new combinations.  Although it is well written, clearly presented and documented, with consistent conclusions and discussions, it would be appropriate to specify the inclusion and exclusion criteria, as well as the study limitation.

Answer: I added a paragraph at the end of the Discussion section to acknowledge all the limitations of the study, including the issues about the VITEK 2 and ETEST methods’ performance assessment.